# Improved Approximation Algorithms for Chromatic and Pseudometric-Weighted Correlation Clustering

**Chenglin Fan**
Department of Computer Science and Engineering
Seoul National University
Seoul, 08826, South Korea
fanchenglin@snu.ac.kr

**Dahoon Lee**
Department of Mathematical Sciences
Seoul National University
Seoul, 08826, South Korea
dahoon46@snu.ac.kr

**Euiwoong Lee**
Computer Science and Engineering Division
University of Michigan
Ann Arbor, MI 48109, USA
euiwoong@umich.edu *

## Abstract

Correlation Clustering (CC) is a foundational problem in unsupervised learning that models binary similarity relations using labeled graphs. While classical CC has been widely studied, many real-world applications involve more nuanced relationships, either multi-class categorical interactions or varying confidence levels in edge labels. To address these, two natural generalizations have been proposed: *Chromatic Correlation Clustering*, which assigns semantic colors to edge labels, and *pseudometric-weighted Correlation Clustering*, which allows edge weights satisfying the triangle inequality. In this paper, we develop improved approximation algorithms for both settings. Our approach leverages LP-based pivoting techniques combined with problem-specific rounding functions. For the pseudometric-weighted correlation clustering problem, we present a tight $\frac{10}{3}$-approximation algorithm, matching the best possible bound achievable within the framework of standard LP relaxation combined with specialized rounding. For the Chromatic Correlation Clustering (CCC) problem, we improve the approximation ratio from the previous best of 2.5 to 2.15, and we establish a lower bound of 2.11 within the same analytical framework, highlighting the near-optimality of our result.

## 1 Introduction

Clustering is a fundamental task in unsupervised learning, where the goal is to partition a set of objects into groups based on their pairwise relationships. One prominent problem in this domain is *Correlation Clustering* (CC) [5], which models binary similarity/dissimilarity between items using an edge-labeled graph: similar pairs are marked with a '$+$' label and dissimilar pairs with a '$-$'. The objective is to partition the nodes to minimize disagreements—i.e., cases where the partitioning contradicts the edge labels. Due to its flexibility in not requiring a predefined number of clusters, CC has been widely utilized in various areas such as detecting communities in networks [17], inferring labels from user interactions [1, 13] and resolving ambiguous entities [28].

---

*Authors are listed in alphabetical order.

However, classic CC models only binary relationships, which is insufficient for many practical applications. For example, in a social network, edges may represent diverse relationship types such as "colleague," "classmate," or "family." To address this limitation, Bonchi et al. [10] introduced the *Chromatic Correlation Clustering* (CCC) problem, which generalizes CC to multi-class categorical settings. In CCC, the input is an edge-colored graph where each color represents a different relationship type. The goal is to cluster the nodes and assign a single color to each cluster such that the number of *disagreements*—edges whose color does not match the cluster's assigned color, or edges that should be separated—is minimized. CCC has wide applications in link classification, entity resolution, and clustering in bioinformatics [10, 3, 30].

In parallel, another important generalization of CC is the *weighted correlation clustering problem*, where edges are associated with weights reflecting the reliability or cost of violating a given label. When weights are unrestricted, obtaining a constant-factor approximation is known to be hard (under the Unique Games Conjecture) [29]. However, when edge weights form a *pseudometric*—i.e., they satisfy the triangle inequality—constant-factor approximations become feasible. This weighted setting more faithfully models scenarios where not all edges are equally trustworthy.

## 1.1  Related Works

The *Correlation Clustering* problem has been widely studied since its introduction [9], and it is known to be APX-hard, leading to efforts to develop approximation algorithms. Early work by Bansal, Blum, and Chawla introduced a constant-factor approximation algorithm [5]. Charikar et al. [15] improved this to a 4-approximation using linear programming. Ailon, Charikar, and Newman then introduced the *Pivot* algorithm [2], which achieved a 3-approximation in linear time. Chawla et al. [16] further improved this to 2.06 using more refined LP-rounding techniques. More recently, researchers have surpassed the 2-approximation barrier. Cohen-Addad, Lee, and Newman [23] used the *Sherali-Adams* hierarchy to develop a $(1.994 + \varepsilon)$-approximation, while Cohen-Addad et al. [22] proposed *preclustering*, which improved the approximation to $(1.73 + \varepsilon)$. The most recent breakthrough by Cao et al. [12] introduced the *cluster LP*, which unifies all known LP relaxations for CC. They show that this can be approximated efficiently using preclustering, achieving a $(1.437 + \varepsilon)$-approximation, the best known guarantee for CC so far. In a more recent work [12], they introduced a new approach to find a feasible solution for the cluster LP in sublinear time.

Chromatic Correlation Clustering is an extension of the classical Correlation Clustering problem, where edge colors represent different types of relationships. Bonchi et al. [10] introduced CCC with a heuristic lacking guarantees. Anava et al. [3] gave a 4-approximation via LP rounding, plus two practical methods: Reduce and Cluster (RC, ratio 11) and Deep Cluster (DC). Klodt et al. [30] showed that Pivot [2] yields a 3-approximation and that RC achieves a 5-approximation. More recently, Xiu et al. [32] developed a 2.5-approximation algorithm for CCC based on a linear programming approach, improving upon the previous best-known ratio. They also introduced a greedy heuristic that achieves strong empirical results.

In modern data analysis, correlation clustering must often be performed under computational constraints such as limited memory or streaming access to data. Consequently, substantial research has focused on crafting clustering algorithms specifically tailored for dynamic, streaming, online, and distributed settings [31, 25, 27, 18, 26, 19, 4, 21, 7, 8, 6, 20, 11].

## 1.2  Our Results

**Our Contributions.**   In this work, we present improved approximation algorithms for both the CCC and pseudometric-weighted CC problems.

- For the *pseudometric-weighted correlation clustering* problem, we develop a refined LP-based pivoting algorithm that achieves a tight $\frac{10}{3}$**-approximation**. We further prove that this approximation factor is *optimal* within the standard LP relaxation framework combined advanced rounding functions.

- For the *Chromatic Correlation Clustering*  problem, we enhance the LP-based method through a new analysis that yields a 2.15**-approximation**, improving upon the previous best bound of 2.5 by Xiu et al. [32]. We also establish a lower bound of 2.11 within the same analytical framework, underscoring the near-optimality of our approach.

Both results are obtained by extending and unifying the triple-based analysis of LP-rounding schemes. Our work improves the theoretical guarantees for two natural and practically motivated generalizations of correlation clustering and contributes new insights into their structural and algorithmic properties.

**Technical Overview.** Our algorithms for both Chromatic Correlation Clustering (CCC) and pseudometric-weighted Correlation Clustering (CC) build on linear programming (LP) relaxations and a unified triple-based rounding framework [16] . Below, we outline the key technical insights:

**Pseudometric-Weighted CC:** The upper bound for the approximation factor $10/3$ is derived using the LP-based Pivot algorithm and a more careful rounding function. For the lower bound, By assuming the existence of an $\alpha$-approximation and analyzing carefully constructed hard instances, the technique derives necessary conditions that any rounding function must satisfy. These conditions expose inherent conflicts, demonstrating that $\alpha$ cannot be arbitrarily small. In particular, the analysis establishes that $\alpha$ must be at least $\frac{10}{3}$. The core idea is to identify instance configurations that induce contradiction between the properties required of the rounding functions, ultimately leading to this lower bound on $\alpha$.

**Chromatic Correlation Clustering (CCC):** Building on the LP formulation introduced by Xiu et al. [32], which jointly encodes fractional cluster membership and color assignments. The decoupling of color assignment from cluster formation, allowing us to preserve color structure without entangling it with clustering decisions. Using a triple-based analysis, we introduce tailored rounding functions—particularly for neutral edges—to better align the rounding behavior with the LP's structure and avoid overcounting. This careful handling of intra-color, conflicting, and neutral edges reduces the approximation factor from 2.5 to 2.15. Our lower bound analysis builds on the general triple-based framework, augmented with structural insights specific to the LP-CCC algorithm and its associated LP solution. We carefully define the cost and LP contribution of each edge type—particularly neutral edges—and construct adversarial instances that expose limitations of any rounding strategy.

**Paper Organization.** The remainder of the paper is structured as follows: **Section 2** introduces the problem formulations and LP relaxations for both pseudometric-weighted and chromatic correlation clustering. **Section 3** presents our approximation algorithms and outlines their design. **Section 4** defines the rounding functions used in the LP-based algorithms. **Section 5** provides a detailed triple-based analysis of the approximation guarantees. We conclude with a summary and discussion in **Section 6**.

## 2 Preliminaries

The *correlation clustering* (CC) problem takes as input a signed undirected graph $G = (V, E = E^+ \uplus E^-)$, where each edge $e = uv \in E$ is assigned a sign '+' or '−', described by $e \in E^+$ or $e \in E^-$. The objective is to find a partition of the nodes such that the number of *disagreements*—i.e., negative edges within the same cluster and positive edges between different clusters—is minimized. In other words, the cost of the clustering $\mathcal{C}$ is as follows:

$$\text{obj}(\mathcal{C}) := \sum_{uv \in E^+} x_{uv} + \sum_{uv \in E^-} (1 - x_{uv}),$$

where $x_{uv} = 0$ indicates that there exists $C \in \mathcal{C}$ such that $u, v \in C$, and $x_{uv} = 1$ otherwise.

CC has a standard LP relaxation leveraging the viewpoint on $x$ as a discrete metric between partitions. Since the $x$ above satisfies the triangle inequality, we can relax the range of $x$ from $\{0, 1\}$ to $[0, 1]$, resulting in the following LP:

$$\text{minimize} \quad \sum_{uv \in E^+} x_{uv} + \sum_{uv \in E^-} (1 - x_{uv}) \tag{CC-LP}$$

$$\text{subject to} \quad x_{uv} + x_{vw} \geq x_{wu}, \tag{1}$$

$$x_{uv} \in [0, 1]. \tag{2}$$

The integrality gap of CC-LP on a complete graph is known to be 2 [15], which indicates that the standard LP-based algorithm cannot obtain a better approximation factor below 2.

## 2.1 Pseudometric-weighted Correlation Clustering

The *weighted Correlation Clustering* problem is a generalization of the classical CC problem in which each edge is associated with a nonnegative violation cost. Specifically, for each edge $uv$ in a complete graph, a weight $w_{uv} \geq 0$ is provided, and violating the edge's label (either '+' or '−') incurs a penalty of $w_{uv}$. This differs from the standard setting, where all violations incur a uniform cost of 1. The weighted variant allows us to encode edge-wise reliability: when $w_{uv}$ is large, it is more reasonable to follow the given label between $u$ and $v$.

However, assuming the *Unique Games Conjecture*, no $O(1)$-approximation algorithm exists for the general weighted case [24]. An exception occurs when the weight function satisfies the triangle inequality, i.e., the weights form a *pseudometric*. In this *pseudometric-weighted* setting, a constant-factor approximation is known [14]. Following the analysis of Charikar and Gao with $L = 2$ yields an approximation factor of $\overline{B}_{\mathrm{HR}} + \frac{1}{3} \leq \frac{4}{3} + 2(L-1) + \frac{1}{3} = 17$, since the second type of charge occurs at most $L - 1 = 1$ time in the charging scheme. The following is a natural LP relaxation of the weighted CC problem, extending (CC-LP):

$$\text{minimize} \quad \sum_{uv \in E^+} w_{uv} \cdot x_{uv} + \sum_{uv \in E^-} w_{uv} \cdot (1 - x_{uv}) \qquad \text{(wCC-LP)}$$

$$\text{subject to} \quad x_{uv} + x_{vw} \geq x_{wu}, \qquad (3)$$

$$x_{uv} \in [0,1]. \qquad (4)$$

Here, the variable $x$ can be viewed as defining a pseudometric over the vertex set, representing the distance between clusters. Since CC on bipartite graphs has an integrality gap of 3 [16], and is a special case of pseudometric-weighted CC, the LP relaxation (wCC-LP) for pseudometric-weighted CC also has an integrality gap of at least 3.

## 2.2 Chromatic Correlation Clustering Problem

The *Chromatic Correlation Clustering* problem is a variant of the classical CC problem in which each cluster is additionally assigned a color [10]. The input includes a complete graph $\left(V, \binom{V}{2}\right)$ and a set of $L$ possible colors, as well as a special color $\gamma$ that denotes that two vertices should not be placed in the same cluster—analogous to a negative ('−') edge in the classical CC setting. When $L = 1$ (i.e., a single cluster color), CCC reduces to the standard CC problem with a complete instance.

The following is a linear programming (LP) relaxation of the CCC problem [10]:

$$\text{minimize} \quad \sum_{\phi(uv) \neq \gamma} x_{uv}^{\phi(uv)} + \sum_{\phi(uv) = \gamma} \sum_{c \in L} (1 - x_{uv}^c) \qquad \text{(CCC-LP)}$$

$$\text{subject to} \quad x_{uv}^c \geq x_u^c, \ x_v^c, \qquad (5)$$

$$x_{uv}^c + x_{vw}^c \geq x_{wu}^c, \qquad (6)$$

$$\sum_{c \in L} x_u^c = |L| - 1, \qquad (7)$$

$$x_u^c, \ x_{uv}^c \in [0,1]. \qquad (8)$$

Here, the variables $x_u^c$ and $x_{uv}^c$ are soft assignments:

- $1 - x_u^c \in [0,1]$ represents the fractional assignment of vertex $u$ to a cluster of color $c$.
- $1 - x_{uv}^c \in [0,1]$ indicates the fractional agreement between vertices $u$ and $v$ under color $c$.

These variables measure the likelihood of vertices or edges being assigned to a color, with $\{1 - x_u^c\}_{c \in L}$ forming a probability distribution over the colors assigned to vertex $u$, subject to constraints (7) and (8).

There is also a geometric interpretation of these variables. Consider $L$ discrete pseudometric spaces $(V_c, d_c)$ where $V_c = \{u_c : u \in V\}$, and vertex $u$ is connected to $u_c$ with a link of length $x_u^c$. Then,

$x_{uv}^c$ represents the *bottleneck distance* between $u$ and $v$, conditioned on traversing the auxiliary connections $u \to u_c$ and $v \to v_c$. This view generalizes the classical CC setting, where the cluster-wise discrete metric can be regarded as a special case of bottleneck distances.

Since CCC generalizes the standard CC problem, the integrality gap of (CCC-LP) is at least as large as that of (CC-LP), which is 2.

## 3 Approximation Algorithm

Building on the LP formulations introduced in the previous sections, we now present approximation algorithms for both pseudometric-weighted CC and CCC settings. LP-PIVOT (Algorithm 1) extends the classical LP-based pivoting method. The set of edges is divided into 3 subsets: $E^+$ and $E^-$ indicate a set of '+' and '−' edges, respectively, while $E^\circ$ indicates a set of edges that always incur a cost regardless of the output. The last subset is involved in the CCC case, as some of the edges might already be misclassified before the execution. $f^+$, $f^-$, and $f^\circ$ are rounding functions, which are explained in Section 4. The time complexity of the algorithm is $O(|V|^2)$.

---

**Algorithm 1** LP-PIVOT

---

**Input:** Complete graph $G = \left(V, \binom{V}{2} = E^+ \uplus E^- \uplus E^\circ\right)$, LP solution $\{x_{uv}\}_{uv \in \binom{V}{2}}$.
**Output:** Clustering $\mathcal{C}$ of $V$.

Pick a pivot $v \in V$ uniformly at random.
Set $C = \{v\}$.
**for** $u \in V \backslash \{v\}$, **do**
    Set $p_{uv}$ as following:
$$p_{uv} = \begin{cases} f^+(x_{uv}), & uv \in E^+; \\ f^-(x_{uv}), & uv \in E^-; \\ f^\circ(x_{uv}), & uv \in E^\circ. \end{cases}$$
    Update $C \leftarrow C \cup \{u\}$ with probability $1 - p_{uv}$.
**end for**
**return** $\mathcal{C} = \{C\} \cup$ LP-PIVOT$(G|_{V \backslash C}, x|_{V \backslash C})$.

---

The algorithm for the pseudometric-weighted CC problem is LP-PIVOT$((V, E^+ \uplus E^- \uplus \emptyset), \{x_{uv}^*\}_{uv \in \binom{V}{2}})$ along with selected rounding functions given by equation (11). The time complexity of the algorithm is dominated by solving the LP, which is polynomial in $|V|$.

The algorithm for the CCC problem is LP-CCC$(G, \phi, x)$ (Algorithm 2) along with the differently selected rounding functions given by equations (12) and (13), which first partitions the vertices according to their LP-derived color distributions, followed by applying the LP-PIVOT algorithm, with edge types partitioned by color. The final clustering is obtained by combining the $|L|$ number of outputs from the LP-PIVOT algorithm. The color pre-classification step requires $O(|V||L|)$ time and the following LP-PIVOT step requires at most $O(|V|^2)$ time in total, which are both dominated by the time complexity of solving (CCC-LP), which is polynomial in $|V|$ and $|L|$.

## 4 Rounding Functions

The effectiveness of the LP-PIVOT and LP-CCC algorithms critically depends on the choice of rounding functions used in the clustering process. Rounding functions $f^+$, $f^-$, $f^\circ : [0, 1] \to [0, 1]$ convert the LP value $x_{uv}$ to the non-selection probability $p_{uv}$ [16]. The sign of the edge $uv$—either '+', '−', or '∘'—determines which rounding function is applied. The sign '∘' indicates that the edge does not belong to $E$. The following natural conditions are imposed on any rounding function $f$:

$$f(0) = 0, \ f(1) = 1; \tag{9}$$
$$x < y \Rightarrow f(x) \le f(y). \tag{10}$$

**Algorithm 2** LP-CCC

---

**Input:** Complete graph $G = \left(V, E = \binom{V}{2}\right)$, color function $\phi : E \to L \cup \{\gamma\}$, LP solution $\{x_u^c\}_{u \in V, c \in L}$ and $\{x_{uv}^c\}_{uv \in E, c \in L}$.
**Output:** Clustering $\mathcal{C}$ of $V$, Coloring function $\Phi : \mathcal{C} \to L$.

Initialize $\mathcal{C} = \emptyset$, $S_c = \emptyset$ for all $c \in L$.
**for** $u \in V$ **do**
    **if** $\exists c \in L$ s.t. $x_u^c < \frac{1}{2}$, **then**
        Update $S_c \leftarrow S_c \cup \{u\}$.
    **else**
        Update $\mathcal{C} \leftarrow \mathcal{C} \cup \{\{u\}\}$.
        Assign $\Phi(\{u\})$ as an arbitrary color.
    **end if**
**end for**
**for** $c \in L$ **do**
    $G_c = (S_c, E_c = E_c^+ \uplus E_c^- \uplus E_c^\circ)$, where $E_c = \binom{S_c}{2}$,
    and $E^+ \uplus E^- \uplus E^\circ$ is defined as a partition by color $c, \gamma, L \backslash \{c\}$ respectively.
    Set $\mathcal{C}_c = \text{LP-PIVOT}(G_c, x^c|_{E_c})$.
    Update $\mathcal{C} \leftarrow \mathcal{C} \cup \mathcal{C}_c$.
    Assign $\Phi(C) = c$ for all $C \in \mathcal{C}_c$.
**end for**
**return** $\mathcal{C}$, $\Phi$.

---

Condition 9 is not only intuitive but also necessary in certain cases, such as ensuring $f^+(0) = 0$ and $f^-(1) = 1$. Other constraints are not required in the proofs of Theorems 1 and 3, which thus provide lower bounds on the approximation factors for *'general'* rounding functions.

**Lemma 1.** *The* LP-PIVOT *algorithm achieves a constant-factor approximation in expectation only if* $f^+(0) = 0$ *and* $f^-(1) = 1$.

*Proof.* We prove it by contradiction on some graph instances.

**Case 1** $f^+(0) = 0$. Consider $G = (V, E = \binom{V}{2} = E \uplus \emptyset \uplus \emptyset)$. The optimal clustering is $\mathcal{C}^* = \{V\}$, satisfying $\text{obj}(\mathcal{C}^*) = 0$, and the optimal LP solution is $x^* \equiv 0$.

Suppose $f^+(0) > 0$. Then $\Pr[\text{LP-PIVOT}(G, 0) \neq \mathcal{C}^*] > 0$. Since $\text{obj}(\mathcal{C}) > 0$ if and only if $\mathcal{C} \neq \mathcal{C}^*$, this leads to a contradiction with the assumption of the expected constant factor approximation.

**Case 2** $f^-(1) = 1$. Consider $G = (V, E = \binom{V}{2} = \emptyset \uplus E \uplus \emptyset)$. The optimal clustering is $\mathcal{C}^* = \{\{v\} : v \in V\}$. The following arguments are similar to Case 1. $\qquad\square$

Different variants of the CC problem may use different rounding functions. In this paper, we provide rounding functions for both the pseudometric-weighted CC and CCC problems.

## 4.1 Pseudometric-weighted Correlation Clustering

We propose the following rounding functions that yield a tight approximation factor:

$$f^+(x) = f^-(x) = \begin{cases} 0, & x < 0.4; \\ \frac{5}{3}x, & 0.4 \leq x < 0.6; \\ 1, & x \geq 0.6. \end{cases} \tag{11}$$

With these functions, the algorithm achieves an expected approximation factor of $10/3$. Moreover, no other rounding function can improve this factor, as shown in Section 5.1.

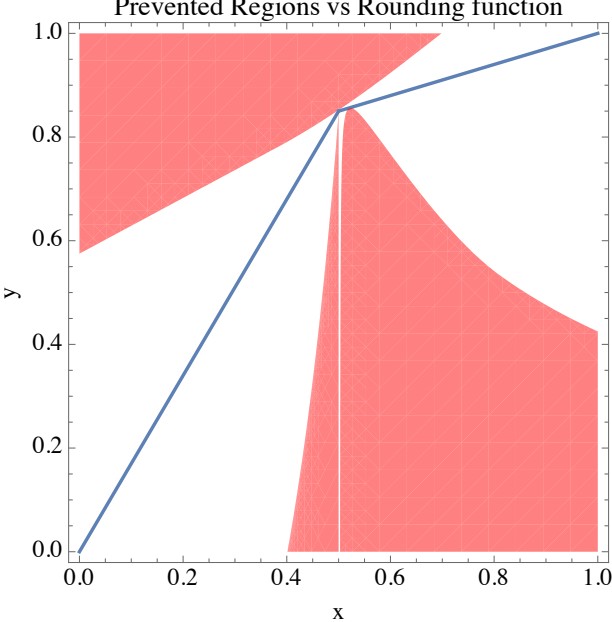

Figure 1: The region where $f^\circ$ violates the $\alpha = 2.15$-approximation for CCC using the proposed $f^\circ$ defined in (13).

## 4.2 Chromatic Correlation Clustering

We further consider the rounding functions $f^+$, $f^-$ from Chawla et al. [16], which yield a 2.06-approximation for classical CC, and introduce a new function $f^\circ$ to handle $\circ$-edges for CCC:

$$f^+(x) = \begin{cases} 0, & x < 0.19; \\ \left(\frac{x - 0.19}{0.5095 - 0.19}\right)^2, & 0.19 \le x < 0.5095; \\ 1, & x \ge 0.5095, \end{cases} \quad f^-(x) = x, \tag{12}$$

$$f^\circ(x) = \begin{cases} 1.7x, & x < 0.5; \\ 0.3x + 0.7, & x \ge 0.5. \end{cases} \tag{13}$$

This function was designed not to intersect with analytic bounds that violate an approximation factor of $\alpha = 2.15$, as illustrated in Figure 1. From Figure 1, $x$ refers to the LP value corresponding to the pivot edge (i.e., one of the endpoints is a pivot vertex) whose color differs from the color under execution of the pivot-based algorithm, while $y$ refers to the corresponding probability of not containing the edge in a single cluster. The plot shows the choice of our $f^\circ$ and the region of $(x, y)$ such that: The triple-based analysis in Section 5 cannot guarantee an $\alpha = 2.15$-approximation if the value $x$ is assigned to the probability of $y$ by the rounding function $f^\circ$.

## 4.3 Comparison with Prior Work

**Pseudometric-weighted CC:** The LP-UMVD-PIVOT algorithm, recently proposed by Charikar and Gao [14] for the Ultrametric Violation Distance (UMVD) problem, follows a pivoting-based rounding strategy applied to an LP relaxation. When the number of distinct pairwise distances between elements, denoted by $L$, is equal to 2, their algorithm can be viewed as a special case of our LP-PIVOT framework. In this setting, the two distances $d_1$ and $d_2$ correspond to the '$-$' and '$+$' labels, respectively, used in our rounding procedure. The rounding functions used are:

$$f^+(x) = f^-(x) = \begin{cases} 0, & x < \alpha; \\ \frac{\max\{x - \alpha\beta, 0\}}{1 - \alpha\beta}, & \alpha \le x \le 1 - \alpha; \\ 1, & x > 1 - \alpha, \end{cases} \quad f^\circ(x) = \begin{cases} 0, & x < \alpha\beta; \\ x, & \alpha\beta \le x \le 1 - \alpha\beta; \\ 1, & x > 1 - \alpha\beta. \end{cases}$$

Here, $\alpha$ and $\beta$ are fixed algorithmic parameters. For the pseudometric-weighted CC problem, they are set to $\alpha = \frac{1}{3}$ and $\beta = 0$, and $f^\circ$ is unused.

With triple-based analysis, this choice yields an approximation factor of 6, as shown in the subsection **??**.

**Chromatic CC:** The LP-based pivoting algorithm by Xiu et al. [32] uses LP values directly as probabilities, corresponding to the following rounding functions:

$$f^+(x) = f^-(x) = f^\circ(x) = x.$$

This setting is known to achieve an approximation factor of 2.5.

# 5 Triple-based Analysis

To complete the analysis of the algorithm, it suffices to show that for every triple of vertices $u, v, w \in V$, the expected cost incurred by the algorithm, denoted $ALG(uvw)$, is at most a factor $\alpha$ times the corresponding LP cost $LP(uvw)$. That is,

$$ALG(uvw) \leq \alpha \cdot LP(uvw).$$

If the inequality holds for every triple, then the total expected cost of the algorithm is at most $\alpha \cdot LP$. To show this, the analysis expresses the expected algorithmic cost and LP cost as averages over all possible pivot choices and vertex triples. Specifically, it defines:

- $e.cost_w(u, v)$: the expected cost of violating constraint $(u, v)$, conditioned on pivot $w$,
- $e.lp_w(u, v)$: the expected LP *charge* of edge $(u, v)$, conditioned on pivot $w$.

Here, LP *charge* is an event that either of the endpoints of the edge is gathered with the pivot vertex, multiplied by the LP value of the edge. Since charging occurs exactly once for each edge, accumulating every charge results in exactly the LP cost.

## 5.1 Pseudometric-weighted Correlation Clustering

In the CC setting, we use the function $\mathcal{C}$, as defined in [16], to measure the gap:

$$\mathcal{C}(x_{uv}, x_{vw}, x_{wu}, p_{uv}, p_{vw}, p_{wu}) = \alpha \cdot LP(uvw) - ALG(uvw),$$

where

$$ALG(uvw) = e.cost_w(uv) + e.cost_u(vw) + e.cost_v(wu),$$
$$LP(uvw) = e.lp_w(uv) + e.lp_u(vw) + e.lp_v(wu),$$

and

$$e.cost_w(u, v) = \begin{cases} p_{uw}(1 - p_{vw}) + (1 - p_{uw})p_{vw}, & uv \in E^+; \\ (1 - p_{uw})(1 - p_{vw}), & uv \in E^-, \end{cases}$$

$$e.lp_w(u, v) = \begin{cases} (1 - p_{uw}p_{vw})x_{uv}, & uv \in E^+; \\ (1 - p_{uw}p_{vw})(1 - x_{uv}), & uv \in E^-. \end{cases}$$

In the weighted CC setting, edge weights further influence the value of $\mathcal{C}$:

$$\mathcal{C}(x_{uv}, x_{vw}, x_{wu}, p_{uv}, p_{vw}, p_{wu}, w_{uv}, w_{vw}, w_{wu}) = \alpha \cdot LP(uvw) - ALG(uvw),$$

with the definition for $e.cost$ and $e.lp$ remains the same; the classical CC corresponds to $(w_{uv}, w_{vw}, w_{wu}) = (1, 1, 1)$.

Under the pseudometric constraint on weights $w$, we can reduce the number of cases to consider in the analysis.

**Lemma 2.** *If $\alpha \cdot LP(uvw) - ALG(uvw) \geq 0$ holds for weight configurations $(w_{uv}, w_{vw}, w_{wu}) \in \{(1, 1, 0), (1, 0, 1), (0, 1, 1)\}$, then the inequality also holds for any configuration $(w_{uv}, w_{vw}, w_{wu})$ satisfying the triangle inequality.*

*Proof.* Let all $x_{uv}$, $x_{vw}$, $x_{wu}$, $p_{uv}$, $p_{vw}$, $p_{wu}$ be fixed. $ALG(uvw)$ and $LP(uvw)$ can be written as

$$ALG(uvw) = w_{uv} \cdot e.cost_w(uv) + w_{vw} \cdot e.cost_u(vw) + w_{wu} \cdot e.cost_v(wu)$$

and

$$LP(uvw) = w_{uv} \cdot e.lp_w(uv) + w_{vw} \cdot e.lp_u(vw) + w_{wu} \cdot e.lp_v(wu).$$

Therefore, the function $\alpha LP(uvw) - ALG(uvw)$ is linear w.r.p. $(w_{uv}, w_{vw}, w_{wu})$.

Since the set of $(w_{uv}, w_{vw}, w_{wu})$ that satisfies the triangle inequality forms a convex cone generated by $(1, 1, 0)$, $(1, 0, 1)$, $(0, 1, 1)$, the function value is nonnegative for all such $(w_{uv}, w_{vw}, w_{wu})$ if and only if the value is nonnegative for $(w_{uv}, w_{vw}, w_{wu}) \in \{(1, 1, 0), (1, 0, 1), (0, 1, 1)\}$. $\qquad\square$

This lemma implies that the algorithm achieves an approximation factor of $\alpha$ if all of the following inequalities are satisfied for every possible configuration on the triangle $uvw$:

$$e.cost_w(uv) + e.cost_u(vw) \leq \alpha \cdot (e.lp_w(uv) + e.lp_u(vw)),$$
$$e.cost_w(uv) + e.cost_v(wu) \leq \alpha \cdot (e.lp_w(uv) + e.lp_v(wu)),$$
$$e.cost_u(vw) + e.cost_v(wu) \leq \alpha \cdot (e.lp_u(vw) + e.lp_v(wu)).$$

We obtain a lower bound on the approximation factor of LP-PIVOT by verifying the feasibility of rounding functions that satisfy the above inequalities. To this end, we analyze several configurations of LP values and edge signs on triangle $uvw$.

In Theorems 1 and 3, the notation '$(a, b, c)$ with $(s_1, s_2, s_3)$' denotes $(x_{uv}, x_{vw}, x_{wu}) = (a, b, c)$, where each edge sign is given by $s_1$, $s_2$, and $s_3$, respectively.

**Theorem 1.** *The lower bound on the approximation factor of* LP-PIVOT *in pseudometric-weighted correlation clustering is* $10/3$*. The proof is deferred to the Appendix.*

Conversely, there exist rounding functions $f^+$, $f^-$ making the approximation factor of LP-PIVOT by $10/3$, providing that the lower bound above is tight.

**Theorem 2.** *The* LP-PIVOT *algorithm with the rounding function defined in equation 11 yields a* $10/3$*-approximation algorithm for pseudometric-weighted CC. The proof is deferred to the Appendix.*

## 5.2   Chromatic Correlation Clustering

We analyze the performance of the LP-CCC algorithm. This algorithm begins by assigning each vertex to its majority color based on the LP solution, followed by a pivot-based clustering routine.

Due to the strict majority condition, any edge not included in $\biguplus E_c$ must have an LP value of at least $1/2$. Thus, the cost incurred by such edges is at most twice their LP contribution [32].

Within each color class $S_c$, corresponding to color $c$, we follow an analysis similar to that of Chawla et al. [16]: edges of color $c$ are treated as positive edges ($E^+$), edges of the adversarial color $\gamma$ as negative edges ($E^-$), and all other edges as neutral ($E^\circ$).

For positive and negative edges, the definitions of $e.cost$ and $e.lp$ remain consistent with those in [16]. The other three cases, particularly those involving neutral edges, require more careful treatment.

Consider a negative edge $uv \in E^-$: the LP value is

$$e.lp_w(u, v) = \sum_{c' \in L} (1 - x_{uv}^{c'}) \geq 1 - x_{uv}^c.$$

For a neutral edge $uv \in E^\circ$, the expected cost arises from the event that $u$ and $v$ are not separated by $w$, i.e., at least one of them shares a cluster with $w$. The expected cost is thus given by the probability that $u$ and $v$ are not simultaneously separated from $w$.

The LP contribution in this case is the product of this probability with $x_{uv}^{\phi(uv)}$, where $\phi(uv) \neq c$ is the color of edge $uv$ in the input. While $x_{uv}^{\phi(uv)}$ is not tied to color $c$, we can still bound it below using

$x_{uv}^c$, $x_{vw}^c$, $x_{wu}^c$ due to LP constraints [32]:

$$x_{uv}^{\phi(uv)} \geq \max\{x_u^{\phi(uv)}, x_v^{\phi(uv)}\} \tag{5}$$

$$\geq \max\left\{\frac{1}{2}, 1 - x_u^c, 1 - x_v^c\right\} \tag{7,8}$$

$$\geq \max\left\{\frac{1}{2}, 1 - x_{uv}^c, 1 - x_{vw}^c, 1 - x_{wu}^c\right\}. \tag{5}$$

Summarizing the results, we express the expected cost and lower bound on the LP value for a fixed pivot $w$ as follows:

$$e.cost_w(u,v) = \begin{cases} p_{uw}(1 - p_{vw}) + (1 - p_{uw})p_{vw}, & uv \in E^+; \\ (1 - p_{uw})(1 - p_{vw}), & uv \in E^-; \\ 1 - p_{uw}p_{vw}, & uv \in E^\circ; \end{cases} \tag{14}$$

$$e.lp_w(u,v) \geq \begin{cases} (1 - p_{uw}p_{vw})x_{uv}^c, & uv \in E^+; \\ (1 - p_{uw}p_{vw})(1 - x_{uv}^c), & uv \in E^-; \\ (1 - p_{uw}p_{vw})\max\left\{\frac{1}{2}, 1 - x_{uv}^c, 1 - x_{vw}^c, 1 - x_{wu}^c\right\}, & uv \in E^\circ. \end{cases} \tag{15}$$

These formulations are central to the analysis. Since $\alpha \cdot LP - ALG$ is always at least the expression obtained from the LP lower bound, we can prove that this bound is nonnegative.

As in Section 5.1, the algorithm achieves an $\alpha$-approximation if the following inequality holds for all triangles $uvw$:

$$e.cost_w(uv) + e.cost_u(vw) + e.cost_v(wu) \leq \alpha \cdot (e.lp_w(uv) + e.lp_u(vw) + e.lp_v(wu)).$$

This inequality leads to the following result on the approximation guarantee for LP-CCC:

**Theorem 3.** *The approximation factor of* LP-CCC *for CCC is bigger than* 2.11*. The proof is deferred to the Appendix.*

Analogous to the classical CC setting—where the lower bound and the approximation ratio of LP-PIVOT differ by less than 0.04 [16]—augmenting the LP rounding with a suitable $f^\circ$ yields the following:

**Theorem 4.** LP-CCC, *using the rounding functions in* (12) *and* (13)*, achieves a* 2.15*-approximation for Chromatic Correlation Clustering. The proof is deferred to the Appendix.*

## 6 Conclusion

In this work, we studied two important variants of correlation clustering: *pseudometric-weighted correlation clustering* and *chromatic correlation clustering* . For both problems, we developed and analyzed specialized rounding functions that are essential for achieving improved approximation guarantees via the LP-PIVOT algorithm.

For the pseudometric-weighted setting, we proposed a piecewise-linear rounding function tailored for the setting that achieves a $10/3$-approximation, and showed that no alternative function within our analytical framework can yield a better factor. For the chromatic correlation clustering variant, we designed a distinct rounding function that respects the constraints imposed by color restrictions and achieves an approximation factor of $2.15$. The function is constructed using a piecewise-linear form and leverages a careful analysis of triple costs.

Overall, our work highlights the importance of designing principled and variant-specific rounding strategies to extend LP-based techniques to structured clustering problems, yielding strong theoretical guarantees.

## Acknowledgment

This work by CF and DL was partially supported by the New Faculty Startup Fund at SNU.

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
