# OpenReview forum: "Improved Approximation Algorithms for Chromatic and Pseudometric-Weighted Correlation Clustering"
_NeurIPS.cc/2025/Conference — NeurIPS 2025 poster_

### Official Review · Reviewer_HMDC · 2025-06-27

**Clarity:** 3
**Significance:** 3
**Originality:** 4
**Rating:** 5
**Confidence:** 4

**Summary:**

The paper studied polynomial-time algorithms for the pseudo-metric correlation clustering and the chromatic correlation clustering problems. The problems are natural generalizations of the classical correlation clustering problems (with (+) and (-) edges and possibly weights).

- In the pseudo-metric correlation clustering problem, our goal is to put vertices of a given graph $G=(V,E)$ to clusters such that the total number of disagreements, defined as the total *weights* of (+) edges crossing clusters and (-) edges inside the same clusters, are minimized. In general, it is impossible to obtain poly-time algorithms for correlation clustering on weighted graphs under plausible hardness assumptions. The paper instead considered the case when the weights satisfy triangle inequalities and obtained a $10/3$-approximation.
- In the chromatic CC problem, we are given an associated color with each edge (or a label of (-)), and our goal is to produce clusters such that the total number of color disagreements and (-) edges inside the same clusters is minimized. For this problem, the paper obtained a $2.15$ approximation, which improved upon the previous 2,5 approximation given by XHTCH [NeurIPS’22].

For both problems, the paper gave lower bounds for the specific algorithms (using the LP integrity gap).

The core techniques for both problems are linear programming (known in the literature) and new rounding algorithms (which are the main novelty of the paper). With fractional random variables, we can obtain the exact answer in polynomial time for the corresponding linear programs for both problems. To produce an integral solution, one of the popular techniques is the pivot-based rounding: we pick a vertex $w$ uniformly at random, and for each neighboring vertex $u$, we add $u$ to the cluster of $w$ based on the variable $x_{uw}$ and the edge label of $(u,w)$. The main technical contributions of the paper are new rounding functions $f^+$, $f^-$, and $f^\text{neutral}$ for different edge types. The paper adopted a region-based analysis to show the upper bound for the approximation algorithm, which appears to follow the same idea as in CMSY [STOC’15].

**Questions:**

Some of the questions were asked when I was flagging the weaknesses. Some additional questions and comments (comments are mostly about presentations):
- Since your lower bound proofs (for both the pseudometric CC and CCC) are based on integrality gaps, I assume one could not hope for better rounding algorithms based on these LPs, am I correct?
- If yes, then do you believe the more recent correlation clustering LPs, e.g., the one in “Understanding the Cluster LP for Correlation Clustering” [STOC’24] will be helpful for further improvements?
- I do not understand Figure 1. Here, what are the x and y axes? What does the figure want to say? I hope it could give some insights for the choice of the intricate constants.
- Notations $E^\circ$ and $f^\circ$ were used before being defined.
- Missing parentheses for equation (13) on the caption of Figure 1.
- In the appendix, you used tables to show that by the choice of $\alpha$, we have $\alpha\cdot LP - ALG \geq 0$. The captions of these tables are “caption” – change this.

**Ethical Concerns:**

["NO or VERY MINOR ethics concerns only"]

**Final Justification:**

The paper is technically solid and contains enough novel results, as I pointed out in my initial review. The author also engaged in the discussion and answered my questions. I'll be happy to see the paper accepted.

**Limitations:**

The paper is theoretical in nature, and I do not see any immediate negative societal impacts.

**Paper Formatting Concerns:**

According to the paper submission instructions, the appendix should be in the same PDF as the main paper.

**Quality:**

3

**Strengths And Weaknesses:**

I read the first 9 pages of the paper to some fair amount of detail, and I briefly looked into the appendix. My general opinion of the paper is positive: I agree that the pseudometrics CC and CCC are important generalizations for the correlation clustering problem with a wide range of potential applications. The algorithms, while using ideas similar to the ones in the literature, also contain non-trivial technical aspects. I unfortunately did not get a time to check the correctness in detail; a spot check found no issues on this front.

Some weaknesses I want to flag:
- The paper assumes access to the LP solution, which might not be very efficient in practice. Maybe the author could comment on how fast could one solve the LPs listed in this paper (a recent STOC’25 paper showed “correlation clustering LP” (not the one listed in this paper) could be solved in $O(n \text{polylog}(n))$ time – can you get similar guarantees?)
- The concerns about efficiency could also be alleviated by some experiments. Unfortunately, the paper does not contain experiments.
- I also think the paper could do a much better job on the *accessibility* front. I’m fairly familiar with the topic, so I don’t find many issues reading the results. However, when reading the analysis, I was confused why the charges on the triangles should be $\sum_{(u,v) \in E^+}(1-p_{uw}p_{vw})x_{uv}+sum_{(u,v) \in E^-}(1-p_{uw}p_{vw})(1-x_{uv})$. In fact, the definition of “charge”  was never introduced in this paper, and I do not think the general audience will be able to follow (I had to read the section in CMSY [STOC’15] to remind myself why it’s the case).

Overall, I’m supportive of the paper and will be happy to see it get accepted.

---

> ### Author Rebuttal · Authors · 2025-07-26
>
> Thanks for your thoughtful review. Please find our responses for each question in the following:
>
> > Since your lower bound proofs (for both the pseudometric CC and CCC) are based on integrality gaps, I assume one could not hope for better rounding algorithms based on these LPs, am I correct?
> >
>
> **Response**: We apologize for causing such confusion. The lower bound provides that the pivot-based algorithm with an arbitrary rounding function cannot bypass that value. The best-known integrality gaps for pseudometric-weighted CC and CCC are still 3 and 2, respectively. However, when it comes to the case of pseudometric-weighted CC, since the selected rounding functions meet the lower bound of 10/3, we hypothesize the integrality gap for the LP as 10/3 and leave it as future work.
>
> > If yes, then do you believe the more recent correlation clustering LPs, e.g., the one in “Understanding the Cluster LP for Correlation Clustering” [STOC’24] will be helpful for further improvements?
> >
>
> **Response**: Thank you for your insightful comment. Indeed, both variants can be easily formulated as variants of cluster LP, which would provide an algorithm with an approximation factor less than 2 by accounting for the cluster-based algorithm as well, as long as the (near-)optimal solution for the LP is provided. The main challenge for using the cluster LP framework for those variants is obtaining a near-optimal solution for those new LPs.
>
> > I do not understand Figure 1. Here, what are the x and y axes? What does the figure want to say? I hope it could give some insights for the choice of the intricate constants.
> >
>
> **Response**: $x$ refers to the LP value corresponding to the pivot edge (i.e., one of the endpoints is a pivot vertex) whose color differs from the color under execution of the pivot-based algorithm, while $y$ refers to the corresponding probability not to contain the edge in a single cluster. The plot shows the choice of our $f^\circ$, and the region of $(x,y)$ such that: The triple-based analysis in Section 5 cannot guarantee an $\alpha=2.15$-approximation if the value $x$ is assigned to the probability of $y$ by the rounding function $f^\circ$.
>
> Overall, the computation of the region acts as a primary test for our construction of $f^\circ$: Once the proposed function doesn’t intersect with the region, we can attempt to validate the approximation factor of $2.15$.
>
> We’ve added the explanation in Section 4.2 regarding the comment.
>
> > Notations $E^\circ$ and $f^\circ$ were used before being defined.
> >
>
> **Response**: Thank you for pointing out the typo. We’ve added a detailed explanation in Section 3 regarding the comment.
>
> > Missing parentheses for equation (13) on the caption of Figure 1.
> >
>
> **Response**: Thank you for pointing out the typo. We’ve revised regarding the comment.
>
> > In the appendix, you used tables to show that by the choice of $\alpha$, we have $\alpha\cdot LP-ALG\geq 0$. The captions of these tables are “caption” – change this.
> >
>
> **Response**: Thank you for pointing out the typo. We’ve revised regarding the comment.

---

> > ### Comment · Reviewer_HMDC · 2025-08-03
> >
> > Thank you for the response! Figure 1 is much clearer to me now.
> >
> > I have a follow-up question for using the new LPs in the STOC'25 paper for pseudo-metric CC and CCC problems: you said that "both variants can be easily formulated as variants of cluster LP", can you give more details about the LP? Also, what is the main barrier to solving that LP? If we are only interested in polynomial-time algorithms, can we simply run some Ellipsoid algorithms?

---

> > > ### Comment · Area_Chair_jnit · 2025-08-06
> > >
> > > Dear Reviewer HMDC,
> > >
> > > I observed that the authors posted new response to your followup questions. Can you please comment on it?
> > >
> > > Regards,
> > >
> > > AC

---

> ### Author Response · Authors · 2025-08-04
>
> Thank you for your follow-up questions.
>
> > Can you give more details about the LP?
> >
>
> **Response:** In the original cluster LP, the variable $z_S\,(S\subseteq V)$ indicated if $S$ exists as an exact cluster in the clustering. This has induced a relation with the variables in the standard LP as $\sum_{S\ni uv}z_S=1-x_{uv}$ and the covering property $\sum_{S\ni u}z_S = 1$.
>
> For the pseudometric-weighted CC, all the variables and constraints for the cluster LP remains the same; the only difference is the objective function is now changed from $\sum_{uv\in E^+}x_{uv}+\sum_{uv\in E^-}(1-x_{uv})$ to $\sum_{uv\in E^+}w_{uv}\cdot x_{uv}+\sum_{uv\in E^-}w_{uv}\cdot (1-x_{uv})$.
>
> For the CCC, on the other hand, the value of $z_S$ is now partitioned as $\sum_{c\in L}z_S^c$, where $z_S^c$ indicates if $S$ exists as an exact cluster **of color c** in the clustering. Therefore, using relations $\sum_{S\ni u}z_S^c=1-x_{u}^c$ and $\sum_{S\ni uv}z_S^c=1-x_{uv}^c$ as well as the covering property $\sum_{S\ni u}\sum_{c\in L}z_S^c=1$, all the constraints from standard LP for CCC can be verified. The objective function is the same as those from standard LP for CCC, since the relation between $z_S$ variables and $x_{uv}$ variables is already developed above.
>
> > Also, what is the main barrier to solving that LP? If we are only interested in polynomial-time algorithms, can we simply run some Ellipsoid algorithms?
> >
>
> **Response:** The main barrier for solving the cluster LP is that the number of variables $z_S$ (or $z_S^c$ for CCC) is exponential to $|V|$. Using general LP solving algorithms such as the Ellipsoid algorithm would thus take $2^{O(|V|)}$ (multiplied by $\text{poly}(|L|)$ for CCC) time, which is undesirable. Therefore, it is important to find an alternative way to (approximately) solve the cluster LP in polynomial time by exploiting properties of the CC problem, which was one of the main discussions in [STOC’25].

---

> > ### Comment · Reviewer_HMDC · 2025-08-07
> >
> > Thank you for the follow-up reply. I now see the main challenge in solving the LPs for pseudo-metric CC and CCC cases. I have no other questions.

---

### Official Review · Reviewer_dZ2e · 2025-07-01

**Clarity:** 3
**Significance:** 2
**Originality:** 2
**Rating:** 4
**Confidence:** 2

**Summary:**

This paper studied two variants of correlation clustering, i.e., pseudometric-weighted correlation clustering and chromatic correlation clustering, and proposed specialized rounding functions for these two problems to achieve improved theoretical approximation guarantees.

**Questions:**

1. What is the time complexity of the proposed algorithm?
2. Can we extend the proposed algorithms to solve other variants of correlation clustering problems and achieve improved approximation guarantees? If no, what are the difficulties?

**Ethical Concerns:**

["NO or VERY MINOR ethics concerns only"]

**Final Justification:**

Thank you for the responses. I agree with other reviewers comments and would maintain my rating.

**Limitations:**

The authors did not explicitly pinpoint the limitations of the work.

**Quality:**

3

**Strengths And Weaknesses:**

S1. The contributions of this work is clearly stated and justified. The comparison with previous work is provided.

S2. The contributions of this work are theoretically solid. Improved approximation guarantees are achieved for Pseudometric-Weighted CC and CCC problems.

W1. The paper is heavy in mathematical notations. There is a lack of intuitive understanding of the rationale of the proposed solution.

W2. No empirical results are provided to validate the theoretical outcome.

W3. Time complexity analysis is not given.

---

> ### Author Rebuttal · Authors · 2025-07-26
>
> Thanks for your thoughtful review. Please find our responses for each question in the following:
>
> > 1. What is the time complexity of the proposed algorithm?
> >
>
> **Response**: Solving the LP takes $\text{poly}(|V|,|L|)$ time (e.g., $\tilde{O}(|V|^{6.5}|L|^{2.5})$ using Vaidya's algorithm, or can be made significantly faster by exploiting the sparsity of the constraint matrix in practice).\
> The following pivot-based algorithm for pseudometric-weighted CC takes $O(|V|^2)$ time, while for CCC takes $O(|V||L|+\sum_{c\in L}|S_c|^2)=O(|V|(|L|+|V|))$ ($S_c$ is a set of vertices such that $c$ is a color of strict majority).
>
> Although solving the LP is the bottleneck, this can be addressed if the $\tilde{O}(2^{\text{poly}(1/\varepsilon)}|V|)$ time algorithm via the multiplicative weight update (MWU) method for solving the cluster LP [STOC’25] is generalizable to each variant.
>
> > 2. Can we extend the proposed algorithms to solve other variants of correlation clustering problems and achieve improved approximation guarantees? If no, what are the difficulties?
> >
>
> **Response**: As long as the problem possesses both a standard LP and a theoretically guaranteed pivot-based algorithm, the technique is still valid to make the approximation factor closer to the known integrality gap.

---

> > ### Comment · Reviewer_dZ2e · 2025-08-04
> >
> > Thank you for the clarification. I would maintain the scores.

---

### Official Review · Reviewer_2HYj · 2025-07-01

**Clarity:** 3
**Significance:** 3
**Originality:** 3
**Rating:** 4
**Confidence:** 4

**Summary:**

The manuscript addresses two variants of the correlation clustering problem. The first variant is clustering correlation with pseudo metric weights. The second variant is chromatic clustering correlation. In both cases the manuscript provides approximation algorithms based on LP relaxations. Both algorithms provide optimal, or near optimal, results (within the LP framework). On the one hand, the novelty and originality of the paper are somewhat limited, in the sense that the methods and arguments are fairly standard, and the authors are able to obtain improved results via a detailed case analysis and very careful selection of certain rounding functions. On the other hand, the theoretical results improve the state of the art, and this is a significant achievement.

**Questions:**

I would be interested to see answers to the weak points, W1--W3, mentioned above.

**Ethical Concerns:**

["NO or VERY MINOR ethics concerns only"]

**Final Justification:**

The paper provides improved results for two variants of the correlation clustering problem but I have raised some concerns about the technical novelty and the practicality of the work. The authors have answered to these questions, but their response has not added any information that could convince me to change my opinion. Thus, I keep my recommendation to "borderline accept". The positive score is for the theoretical improvements, and the "borderline" for the above mentioned weaknesses.

**Limitations:**

I did not find much discussion on limitations. My overall impression is that the authors do not make a significant effort to offer a critical assessment of their work along different dimensions.

**Paper Formatting Concerns:**

I do not have concerns about format.

**Quality:**

3

**Strengths And Weaknesses:**

Strengths:
S1. The problem of correlation clustering and its variants have been studied extensively in the literature of theoretical computer science. The current manuscript makes a good improvement of the state of the art results, for the approximation ratio of the problems under study.
Weaknesses:
W1. The methods and ideas are not so novel. The improvement has been possible via a detailed case analysis and careful selection of rounding functions. The resulting algorithm may give improved quality guarantee but it is not so novel and not so elegant.
W2. The methods rely on LP relaxation, which is not so practical. Overall, this is a theory paper and there are no experiments.
W3. In terms of presentation, it is not always so clear when the authors refer to the problem where all the edges are present (clique) and when the edges may be missing (no + or - signs, or no color labels).

---

> ### Author Rebuttal · Authors · 2025-07-26
>
> Thanks for your thoughtful review. Please find our responses for each question in the following:
>
> > The methods and ideas are not so novel. The improvement has been possible via a detailed case analysis and careful selection of rounding functions. The resulting algorithm may give improved quality guarantee but it is not so novel and not so elegant.
> >
>
> **Response**: This paper differs from previous papers in the following ways:
>
> 1. To the best of our knowledge, this paper presents the earliest theoretical inspection particularly on the pseudometric-weighted CC problem, whose significance is discussed in Section 2.1.\
> Moreover, since the lower bound of the approximation factor for the pivot-based algorithm is a fractional value of 10/3 and is achievable (which is quite unusual), it can also be hypothesized that the integrality gap of the pseudometric-weighted standard LP is also 10/3. This is slightly higher than the integrality gap of 3 for the easier problem: A CC whose presenting edges form a k-partite graph. Providing such an integrality gap of 10/3 would indicate that there are more features from pseudometric compared to ultrapseudometric (i.e., a pseudometric retaining cluster-like features) as a weight that affects the hardness of the LP or the problem.
>
> 2. This paper also emphasizes the applicability and the effectiveness of the rounding technique that was explicitly used only for the original CC before (as far as we know). Whenever the triple-based analysis is involved, one can think of fitting the rounding functions, which is simple yet likely to be effective, as in the result of the paper.\
> Despite its simplicity, it is also important to check the extent of its applicability and to check if the overall procedure is analyzable. For example, while designing the algorithm for CCC, we had to define an additional sign of ‘$\circ$’ with the corresponding rounding function; while analyzing the performance for pseudometric-weighted CC, the ideas that the tuple $(w_{uv},w_{vw},w_{wu})$ satisfying triangle inequality forms a convex cone and that $\alpha\cdot LP-ALG$ is affine with respect to the tuple were crucial in reducing uncountable cases for analysis to only 3 cases consisting extremal rays of the cone.
>
> > The methods rely on LP relaxation, which is not so practical. Overall, this is a theory paper and there are no experiments.
> >
>
> **Response**: Although solving the LP might be a limitation on the practicality, recent work on solving cluster LP implies that the near-optimal solution for the LP can be constructed in $\tilde{O}(2^{\text{poly}(1/\varepsilon)}|V|)$ time using multiplicative weight update (MWU) method if the framework is generalizable to each variant.\
> Moreover, since the state-of-the-art algorithm for CC uses both pivot-based and cluster-based algorithms, the proposed rounding functions can behave as a good initial choice of rounding functions for the pivot-based algorithm in the mixed algorithm for variants.
>
> > In terms of presentation, it is not always so clear when the authors refer to the problem where all the edges are present (clique) and when the edges may be missing (no + or - signs, or no color labels).
> >
>
> **Response**: Thank you for pointing out the ambiguity. In every case, all the edges are present; missing edges can only be encoded as arbitrary-signed edges with zero weight for pseudometric-weighted CC. We’ve made a clarification regarding the comment.

---

> > ### Comment · Reviewer_2HYj · 2025-08-03
> >
> > Thank you for the clarifications. After reading the other reviews and the author responses I am leaning towards maintaining my assessment about this submission.

---

### Decision · Program_Chairs · 2025-09-17

**Decision:**

Accept (poster)

**Comment:**

This paper studies two variants of correlation clustering: chromatic version and weighted version. The main results are constant approximation algorithms that run in polynomial time. The variants of the problems are relevant to ML, the result is theoretically solid, and the techniques are based on sophisticated LP-rounding framework. Both results improve existing approximation ratios. The main weakness is that the algorithm can run in high-degree polynomial which may hurt the scalability and applicability in practice.

Overall, this is still a solid contribution that may lead to further studies. The reviews are generally positive, and consensus is basically reached during rebuttal. I recommend to accept the paper.